# Social Environmental Factors Related to Resuming Driving after Brain Injury: A Multicenter Retrospective Cohort Study

**DOI:** 10.3390/healthcare9111469

**Published:** 2021-10-29

**Authors:** Mamiko Sato, Yasutaka Kobayashi, Kazuki Fujita, Masahito Hitosugi

**Affiliations:** 1Department of Rehabilitation Medicine, Fukui General Hospital, Fukui 910-8561, Japan; 2Graduate School of Health Science, Fukui Health Science University, Fukui 910-3190, Japan; yasutaka_k@fukui-hsu.ac.jp (Y.K.); k.fujita@fukui-hsu.ac.jp (K.F.); 3Department of Legal Medicine, Shiga University of Medical Science, Shiga 520-2192, Japan; hitosugi@belle.shiga-med.ac.jp

**Keywords:** resuming driving, social environmental factors, International Classification of Functioning (ICF), brain injury

## Abstract

Many patients resume driving after brain injury regardless of their ability to drive safely. Predictors for resuming driving in terms of actual resumption status and environmental factors are unclear. We evaluated the reasons for resuming driving after brain injury and examined whether social environmental factors are useful predictors of resuming driving. This retrospective cohort study was based on a multicenter questionnaire survey at least 18 months after discharge of brain injury patients with rehabilitation. A total of 206 brain injury patients (cerebrovascular disease and traumatic brain injury) were included in the study, which was conducted according to the International Classification of Functioning (ICF) items using log-binominal regression analysis, evaluating social environmental factors as associated factors of resuming driving after brain injury. Social environmental factors, inadequate public transport (risk ratio (RR), 1.38), and no alternative driver (RR, 1.53) were included as significant independent associated factors. We found that models using ICF categories were effective for investigating factors associated with resuming driving in patients after brain injury and significant association between resuming driving and social environmental factors. Therefore, social environmental factors should be considered when predicting driving resumption in patients after brain injury, which may lead to better counseling and environmental adjustment.

## 1. Introduction

Predicting the likelihood of driving resumption in patients after brain injury is a major concern for patients and their families [1,2,3]. Motor, visual, cognitive, and perceptual problems after brain injury affect driving ability [3,4,5,6] and make it difficult to resume driving. Studies have shown that 30–61% of patients return to driving after brain injury [1,7,8,9,10].

Many previous studies have focused on the physical and cognitive problems that affect fitness to drive after brain injury [11,12,13]. The current Road Traffic Law in Japan requires judgment of driving ability based on the presence or absence of cognitive, predictive, judgmental, or manipulative ability, and physicians must use their expertise to comprehensively diagnose the impact on driving and provide guidance to patients and their families. It has been reported that one-third of stroke patients resume driving without receiving advice on driving from medical professionals [7]. Therefore, while it is important to judge fitness to drive from a medical point of view, it is equally important to assess the background and conditions of patients who resume driving. In Italy, a scientometric analysis of driving simulation review was conducted [14], and a factor structure to access driving behavior and attitudes towards traffic safety and self-regulation in driving has been developed [15]. Only a few studies have examined factors associated with resuming driving after brain injury [3,16,17,18]. These studies mainly focused on physical structure, and the Barthel index score and functional independence measure (FIM) were reported as useful predictors of resuming driving. However, Perrier [16] applied the framework of the International Classification of Functioning (ICF) to driving and stated that there are many components to resuming driving, including environmental and individual factors.

The ICF model is a classification of health and health-related conditions that was developed by the World Health Organization (WHO) and published in 2001 [18,19]. This model describes the interaction between “body functions and structures”, “activities”, “participation”, “health condition”, “environmental factors”, and “personal factors”. According to the ICF concept, resuming driving corresponds to “activity”. Therefore, all relevant factors across each element of ICF should be considered for driving. When previously reported factors that affect resuming driving are classified into each element, “body functions and structures” was the most common factor and was evaluated by many items, such as Barthel index score [3], mini-mental state examination [16], cognitive items of FIM, lower extremity mobility index score [17], Fugl-Meyer Assessment score, and National Institutes of Health Stroke Scale [18]. Stroke impact scale and type of stroke [16] were reported as evaluation tools for the factor of “health condition” and pre-stroke driving frequency and marital status [3] were reported as good evaluation items for “personal factors”. Doucet [20] examined “participation” and reported that patients who resumed driving were more likely to return to work, and showed a positive correlation between time to re-employment and time to resuming driving. However, a Canadian study [2] investigating “environmental factors” reported differences in the use of transport between drivers and non-drivers after stroke, although no other studies have shown the effect of “environmental factors” on resuming driving. Therefore, the present study investigated whether considering factors based on the concepts of ICF are good predictors of resuming driving after brain injury and how “environmental factors” contribute to resuming driving using a multicenter questionnaire survey.

## 2. Materials and Methods

### 2.1. Study Population

This retrospective, multicenter cohort study was conducted between April 2013 and April 2018 in four hospitals with a convalescent rehabilitation ward separate from the acute care ward (Fukui General Hospital, Shimada Hospital, Kimura Hospital, Harue Hospital) in Fukui, Japan. Convalescent rehabilitation wards were the main system of inpatient rehabilitation facility introduced in Japan in 2000. Patients requiring assistance with activities of daily living following treatment in an acute care hospital for diseases such as stroke, traumatic brain injury, spinal cord injury, acute neurological diseases, and fracture can be admitted to convalescent rehabilitation wards. A postal questionnaire survey was conducted to assess the social environment surrounding driving and the resumption of driving 18 months after discharge from convalescent rehabilitation wards. The questionnaire asked respondents to choose from the following three options. (1) resumed driving, (2) resumed driving but then stopped, or (3) did not resume driving. Those who stop driving after resuming driving for some reason, such as illness or injury, were also included in the “resumed driving” group based on the interpretation that they had resumed driving once after the brain injury. A caregiver was allowed to write the questionnaire on behalf of the patient if the patient was unable to complete it themselves. The inclusion criteria were as follows: (1) age ≥ 18 years; (2) patients with cerebrovascular disease or traumatic brain injury who received rehabilitation care in a convalescent rehabilitation ward; and (3) consent and information were obtained for the postal questionnaire. The exclusion criteria were as follows: (1) age ≥ 90 years (in accordance with the rules of driving education in our hospital); (2) dementia; (3) visual impairment; and (4) significant physical impairment (motor items of FIM < 60 points at time of discharge). Questionnaires were sent to 507 patients according to the inclusion criteria and responses were received from 232 patients (45.8% response rate). Among these 232 patients, 26 who did not drive prior to brain injury were excluded and a final total of 206 cases were included in the study. The results of the questionnaire survey 18 months after brain injury showed there were 120 patients in the “resumed driving” group (80.0% male) and 86 patients in the “did not resumed driving” group (77.9% males). The resuming driving rate was 58.3% (Figure 1).

The mean age was significantly lower in the “resumed driving” group (60.2 ± 13.1 years) than in the “did not resumed driving” group (65.9 ± 11.2 years) (*p* = 0.001). Table 1 shows the general characteristics of the subjects and the differences between the variable for the “resumed driving” group and the “did not resumed driving” group.

The study was conducted in accordance with the principles of the Declaration of Helsinki, and written informed consent was obtained from all patients prior to inclusion in the study. The study was approved by the Ethical Review Committee of Nittazuka Medical Welfare Center (approval no. Nittazuka Ethics 2019–2027).

### 2.2. Data Collection

Data regarding age, sex, and medical history of brain injury (diagnosis, medication, and neuropsychological test results) were extracted from the participants’ hospital records. Resuming driving was fitted to the model according to the ICF categories of “body functions and structures”, “activities”, “participation”, “health condition”, and “environmental factors”.

For “body functions and structures”, FIM, which reflects physical and cognitive functions, was found to be a relevant factor in the evaluation of resuming driving after brain injury [17,21,22]. FIM was selected as an evaluation item as it has the most data.

For “health condition”, recurrent stroke and epilepsy are considered important according to the Japanese Road Traffic Law’s operational criteria for the admissibility of licenses for specific diseases. In many countries, it is recommended that patients should refrain from driving for one month after a stroke in terms of the risk of recurrence [23,24]. Furthermore, a previous study [25] advocated a period of driving prohibition after an epileptic seizure in consideration of the risk of recurrence. Therefore, the present study investigated the history of stroke and the prevalence of epilepsy.

For “participation”, a study conducted in India [26] concluded that unemployment was a predictor for not being able to resume driving after a stroke and had a significantly negative impact on a person’s social life. As the main aim of the present study was to identify predictors of resuming driving, we examined the relationship between employment status prior to brain injury, rather than return to work status, surveyed in the questionnaire.

For “environmental factors”, Finestone et al. [2] reported that people who resumed driving after a stroke were more likely to rely on friends, family, public transport, and taxis than those who did not resume driving. Kendra [27] reported that elderly drivers with greater impairment to mode of transport showed a statistically significantly increased likelihood of resuming driving. Another study in elderly patients reported [28] that those with a limited social network were more likely to resume driving, whereas adequate support from family and friends was linked with driving cessation. On the ICF documentation, environmental factors are delineated as social and physical factors. In this study, we focused on the social environment and investigated the availability of public transport and the presence of an alternative driver.

Age and sex were included as “personal factors”, whereas other items were excluded from the survey as they were difficult to collect in the questionnaire from the viewpoint of personal information protection.

Finally, we incorporated age and sex into the model. Multivariate analysis was conducted using the dependent variable of whether or not the patient had resumed driving and independent variables of age, sex, motor FIM at admission, cognitive FIM at admission, risk of recurrence or seizure, traffic in the resident’s area, presence of alternative driver, and working status prior to brain injury.

### 2.3. Data Analysis

R and Stata 17 (RightStone Corp, Plano, TX, USA) were used for all statistical analyses. General characteristics of participants and differences in variables between the “resumed driving” and “did not resumed driving” groups were identified using chi-square or unpaired *t* tests or Mann–Whitney U test. Log binominal regression analysis was performed using Stata 17 to identify predictors of resuming driving after brain injury. Correlations and multicollinearity between variables were assessed by tolerance and variance inflation factors. Predictors were estimated using risk ratios (RRs) and 95% confidence intervals (CIs). *p*-values < 0.05 were considered to indicate statistical significance.

## 3. Results

### 3.1. Resuming Driving Status and Each Component of the ICF

#### 3.1.1. Body Functions and Structures

The mean ± SD of motor and cognitive items were significantly higher in the “resumed driving” group than in the “did not resumed driving” group (70.8 ± 18.5 vs. 55.6 ± 22.9 and 30.3 ± 5.8 vs. 25.8 ± 7.9; *p* = 0.04 and *p* = 0.003, respectively). Total FIM was also significantly higher in the “resumed driving” group (101.1 ± 20.6) than in the “did not resumed driving” group (81.6 ± 26.9) (*p* = 0.008).

#### 3.1.2. Health Condition

A total of 54 patients (26.2%) were at risk of recurrence or seizure. The proportion was significantly lower in the “resumed driving” group (20.0%) than in the “did not resumed driving” group (34.9%) (*p* = 0.024).

#### 3.1.3. Participation

A total of 156 patients (75.7%) were employed prior to brain injury. The proportion was significantly higher in the “resumed driving” group (80.8%) than in the “did not resumed driving” group (68.6%) (*p* = 0.049).

#### 3.1.4. Environmental Factors

A total of 124 patients (60.2%) were able to use public transport in their area of residence. The proportion was significantly lower in the “resumed driving” group (29.4%) than in the “did not resumed driving” group (52.4%) (*p* < 0.001). Furthermore, 155 patients (75.7%) had someone to act as an alternative driver. The proportion was significantly lower in the “resumed driving” group (70.7%) than in the “did not resumed driving” group (86.9%) (*p* = 0.009).

### 3.2. Log-Binominal Regression Analysis

Table 2 shows the RRs and CIs for the log-binominal regression model. Age, sex, risk of recurrence or seizure, employment before onset, inadequate public transport, and no alternative drivers showed statistical significance. The RRs were 1.38 (95% Cl, 1.08–1.79; *p* = 0.016) for inadequate public transport, and 1.53 (95% Cl, 1.14–2.04; *p* = 0.004) for no alternative drivers.

## 4. Discussion

The present study examined the current status of resuming driving after brain injury in Japan and found that approximately 58% of brain injury patients resumed driving at the time of evaluation 18 months after onset, which is consistent with the findings of other studies [3,16,17,23,29]. The results of the present study suggested that use of the ICF model was a good predictor of driving resumption after brain injury. In addition, multivariate analysis showed that there was a significant association between resuming driving and social environmental factors, such as inadequate public transport (RR, 1.38) and no alternative drivers (RR, 1.53). It is important to note that the present study indicates that social environmental factors are associated with resuming driving after brain injury, independently of physical and mental functioning, which is often the focus of attention.

In the present study, the independent variables used in the multivariate analysis to adjust for confounding factors affecting driving resumption were in line with each factor of the ICF concept, each using available items reported in previous studies. The ICF framework proposed by the WHO could become a standard for disabling language that focuses on how people live with their condition [19,21]. In addition to health status and physical and mental functions, environmental and personal factors should be considered when applying the concept of ICF to driving as they interact with each other.

The results of the present study suggests that social environmental factors may be independent and important in predicting the resumption of driving after brain injury. Several previous studies [25,30,31,32,33,34] including environmental factors have also reported driving cessation among elderly drivers. A Korean study [31] showed that residential area, which is an environmental factor, was a strong predictor of driving cessation in the elderly (OR, 2.21; CI, 1.86–2.62). Finestone [2] examined the relationship between resuming driving after stroke and environmental factors, and found that people who resumed driving relied more on family and friends on a regular basis, whereas non-drivers were much more dependent on family, friends, public transport, and taxis. However, their study was conducted using drivers who underwent a medical driving evaluation. In contrast, the results of the present study included all patients after brain injury, regardless of their aptitude for driving, as the purpose of the study was to identify predictors of resuming driving.

While it is very important to know if a patient is safe to drive, it is also necessary to predict early on which patients may attempt to drive. We used inpatient assessment items at the time of admission rather than discharge to focus on factors that are predictive at an early stage. Most brain injury survivors do not receive a driving evaluation or advice on resuming driving [16], although providing education on resuming driving has been shown to be strongly associated with resuming driving [29]. It is meaningful to know in advance during the early stages of hospitalization which patients are likely to resume driving when implementing a systematic education program for resuming driving. In other words, it is important to bear in mind that patients with no alternative means of transport or alternative drivers are more likely to resume driving, and rehabilitation should guide the safety of transport to maintain activity.

Restricting individual activities of brain injury patients to maintain social safety is straightforward; however, it is difficult to take compensatory measures as it may limit individual social participation. It is important that medical workers focus on driving as an activity in brain injury patients and consider the significant impact of traffic and family environment to help them to resume driving more safely. Driving education is one approach to achieve this, although it may be necessary for the community to develop a means of transport that is friendly to patients with brain injuries.

The present study has several limitations. First, we were unable to evaluate the adequate cognitive functions that affect driving resumption, such as unilateral spatial neglect. Second, in terms of the local transport environment, driving resumption is influenced by whether the public transport system is well developed. Fukui in Japan is not an urban area wherein public transport would be easily available; rather, it is a rural area with relatively scarce public transport. This means that the local environment is not necessarily representative of the rest of the country, and this should be taken into account. Third, with regard to the study population, most of the subjects had cerebrovascular disease, whereas only 13 had traumatic brain injury; the inclusion of these subject may have biased the results, as the nature of the damage from traumatic brain injury is likely to be broader than that of cerebrovascular disease. In addition, the time at which the driving resumption was investigated varied from patient to patient and included cases who resumed driving but quit midway and, conversely, cases who resumed driving long after the onset of disease. Therefore, further studies should be conducted in areas with different public transport environments, with a uniform study population and time frame for evaluation.

## 5. Conclusions

Our results indicate that the influence of social environmental factors should be considered in the evaluation resuming driving after brain injury. In brain injury patients who received inpatient rehabilitation services, inadequate public transportation and no alternative driver was an independent factor associated with resuming driving after brain injury, with other associations suggested for age, sex, risk of recurrence and seizure, and employment before onset. Clinicians may be able to screen patients who need to resume driving as early as inpatient rehabilitation by considering local transportation and family environment surrounding driving. This model may allow healthcare providers to provide better counseling to brain injury patients and their families about resuming driving and focus their efforts on education and environmental adjustments for resuming driving.

## Figures and Tables

**Figure 1 healthcare-09-01469-f001:**
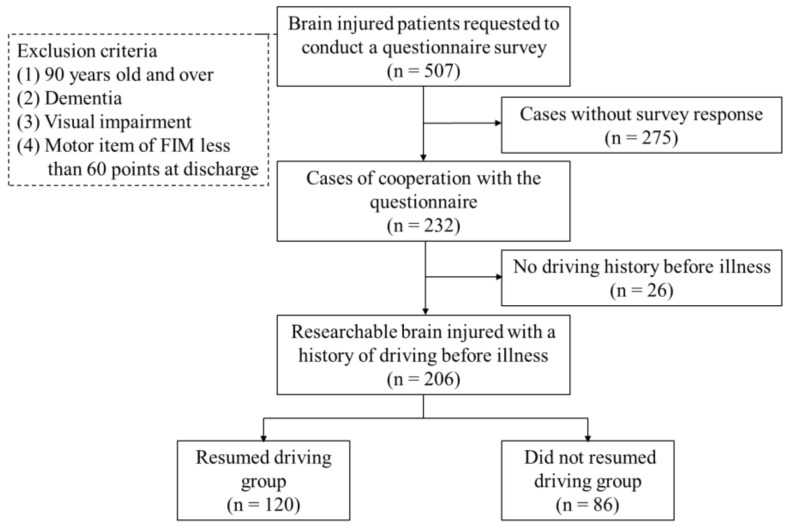
Flowchart of the study selection.

**Table 1 healthcare-09-01469-t001:** General characteristics of the study participants and the differences in variables between the “resumed driving” group and the “did not resumed driving” group.

Characteristic		Resumed Driving Group (*n* = 120)	Did Not Resumed Driving Group (*n* = 86)	*p*-Value
Age (years) mean ± SD		65.9 ± 11.2	60.2 ± 13.1	0.001
Sex [number (%)]	Male	96 (80.0)	67 (77.9)	0.84
	Female	24 (20.0)	9 (22.1)	
Diagnosis [number (%)]	Cerebral infarction	64 (53.3)	48 (55.8)	0.100
	Cerebral hemorrhage	40 (33.3)	30 (34.9)	
	Subarachnoid hemorrhage	9 (7.5)	2 (2.3)	
	Traumatic brain injury	7 (5.8)	6 (7.0)	
FIM (points) mean ± SD	Total	101.1 ± 0.6	81.6 ± 26.9	0.008
	Motor item	70.8 ± 18.5	55.6 ± 22.7	0.040
	Cognitive item	30.3 ± 5.8	25.8 ± 7.9	0.003
Risk of recurrence or seizure [number (%)]	Yes	24 (20.0)	30 (34.9)	0.016
	No	96 (80.0)	56 (65.1)	
Employment before onset [number (%)]	Yes	97 (82.2)	59 (70.2)	0.049
	No	21 (17.8)	25 (29.8)	
Inadequate public transport [number (%)]	Yes	84 (70.6)	40 (47.6)	<0.001
	No	35 (29.4)	44 (52.4)	
No alternative driver [number (%)]	Yes	34 (29.3)	11 (13.1)	0.009
	No	82 (70.7)	72 (86.9)	

**Table 2 healthcare-09-01469-t002:** Log-binominal analysis of factors predicting resuming driving after brain injury.

	SE	RR	95% CI	*p*-Value
Age	0.003	0.98	0.97–0.98	<0.001
Sex	0.119	0.65	0.45–0.93	0.019
Motor item of FIM	0.002	1.00	1.00–1.01	0.120
Cognitive item of FIM	0.009	1.02	1.00–1.03	0.068
Risk of recurrent or seizures	0.103	0.6	0.47–0.88	0.006
Employment before onset	0.925	0.78	0.62–0.99	0.038
Inadequate public transport	0.184	1.38	1.06–1.79	0.016
No alternative driver	0.225	1.53	1.14–2.04	0.004

CI, confidence interval; FIM, functional independence measure; RR, risk ratio; SE, standard error.

## Data Availability

Not applicable.

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
