# Peer review of "Social Environmental Factors Related to Resuming Driving after Brain Injury: A Multicenter Retrospective Cohort Study"

_healthcare, 2021, doi:10.3390/healthcare9111469_

Round 1

Reviewer 1 Report

Here, the authors describe a retrospective cohort study evaluating the characteristics of returning to driving 18 months following discharge for a brain injury, predominantly stroke and other haemorrhage, among a sample of 206 patients. The authors apply the WHO International Classification of Functioning framework to assess potential determinants of returning to driving. Alongside functional independence measures, age, and risk of recurrence, returning to driving was most strongly determined by patients having inadequate access to public transport and a lack of having an alternative driver. The authors then conclude such factors should be considered in predicting return to driving in such patients. I have some various comments which I hope the authors will consider.

While the authors state that this is the first study to show that social environmental factors are associated with returning to driving amongst patients following a brain injury, this is rather intuitive and expected. Regardless of their being sufficiently able to drive, a person who regards themselves as functional and who has no alternatives, either public transport or having someone else drive for them, will be more likely to do so. This is the whole reason for driving under the influence restrictions. I have to imagine that on discharge patients are advised about potential risks for their driving (reduced function, risk of relapse) and recommendations for alternatives suggested. While it is useful to have a specific quantification of the measure of association for these relationships, it is not especially novel. Indeed, the authors cite the Moon and Finestone studies in the Discussion, which show that residential area (a proxy for proximity to public transport and other services) and having greater reliance on family and friends (alternative drivers) were associated with returning to driving. Thus, I would suggest instead of framing these results as being novel, discuss how these results both replicate the Moon and Finestone study findings but importantly show each of these factors as being independent of one another and also of measures of functional independence.

The classification of access to public transport and to an alternative driver as environmental factors is not ideal. On review of the ICF documentation, they delineate environmental factors as social and physical. I would argue that these two factors need to be clearly described as social environmental, as just calling them environmental may lead readers, and did indeed lead this reviewer, to believe factors like exposure to environmental features like sun and pollution were what was under consideration. Accordingly, please use this terminology in the title, abstract, and body text.

Please reduce the number of abbreviations. It is not helpful to abbreviate things so much, or even to make abbreviations of abbreviations (RDG, NRDG, for example). I would suggest RD could just be spelled out as this is your primary outcome of interest.

Please introduce earlier, both in the Abstract and the body text, that this was a study of returning to driving 18 months after discharge following the brain injury hospitalisation. This only becomes apparent at section 2.2 but is relevant much earlier.

The inclusion criterion that patients be able to complete a questionnaire does potentially bias the sample to a more functional population, as the patients who couldn’t complete the questionnaire themselves would presumably be of much lower FIMs and less likely to attempt driving. It would have been preferable to have the option for the questionnaire having been completed by a spouse or other contact if the patient was not themselves able. Some acknowledgement of this limitation should be made.

The fact that most brain injury patients were stroke or haemorrhage, and only 13 traumatic brain injury, might make it worthwhile to examine whether exclusion of the TBI materially changed results. I suggest this just because the nature of the damage from TBI is likely broader in area than a stroke or aneurism might realise. If there is no difference, this can just be stated in text.

Japan has lots of public transport, which people might be more disposed to use instead of driving than areas with less public transport. Please comment on this.

Given the study design, I might suggest the authors use log-binomial rather than logistic regression, which would enable estimation of risk ratios. Logistic regression outside of a case-control study can overestimate the magnitude of associations.

Other comments:

  • Add sample size to the Abstract.
  • I would suggest enumeration of the brain injury types (CI, CH, SAH, TBI) to the Abstract, spelled out.
  • ORs and 95% CIs can just be to two decimal places.
  • I’m unclear why patients aged 90+ were excluded? Please clarify.
  • Regarding how driving status was queried, was this just a yes/no or were there other options?
  • Regarding the analysis software, maybe just say R rather than EZR? EZR is evidently just a package accessible in CRAN and thus works via R.
  • In Table 1, suggest presentation of di/polychotomous variables should just be n (%) as is done later in the table, but not for sex or diagnosis.
  • Table 1 needs definition of the abbreviations for diagnosis types. These appear in Table 2 but I think that’s a mistake.
  • Suggest regarding the risk of disease at line 159, this should be risk of recurrence.
  • Throughout text, please don’t use term, prevalence. Use frequency or proportion.
  • In section 3.2, it is not necessary to reiterate measures of association for all factors in Table 2.
  • Also in section 3.2, the area under the curve should be 0.628, not 0.805 which is just the upper bound of the CI.
  • In Table 2, it is not necessary to present the log(OR) column.
  • In Discussion, don’t say confirmed. Also, I might suggest not using the term, predictors.
  • As previous, the Soon study looked at residential area as a characteristic of returning to driving. Please specify what exactly this term means. Is it just urban/rural or what?

Author Response

 The letter are point-by-point responses to the comments raised by the reviewers.

We would like to take this opportunity to express our sincere thanks to the reviewers who identified areas of the manuscript that needed corrections or modification. We would like also to thank you for allowing us to resubmit a revised copy of the manuscript.

We hope that the revised manuscript is accepted for publication in Healthcare.

Reviewer 2 Report

The manuscript entitled  "Environmental factors related to resuming driving after brain injury: a multicenter retrospective cohort study" is interesting and well written, bus some changes are suggested.

Please add at the end of the abstract a take home message.

 Please see Spano et al., 2019 (10.3389/fpsyg.2019.00368), Caffò et al., 2020 (10.3389/fpsyg.2020.00917)
to improve the introduction of the ms.

Please add the power analysis in order to compute the sample size, in the study population. Please move from the results to the study population section the final sample size, the characteristics of the sample and Figure 1.

Please add the future direction of the study in the discussion section

Author Response

This letter are point-by-point responses to the comments raised by the reviewers.

We would like to take this opportunity to express our sincere thanks to the reviewers who identified areas of the manuscript that needed corrections or modification. We would like also to thank you for allowing us to resubmit a revised copy of the manuscript.

We hope that the revised manuscript is accepted for publication in Healthcare.

Responses to Reviewer 2

Please note that, in Table 1, the number and proportion of people in the “resumed” group (percentage) for the “inadequate public transformation” and “inadequate public transport” item was reversed. This error has been amended in the revised manuscript. In addition, we have had the manuscript proofread by a native English speaker and have made minor corrections, such as spelling errors, other than those pointed out.

Please add at the end of the abstract a take home message.

As you point out, it is important to be clear about what you want your message to be.We have added a take-home message to the conclusion and at the end of the Abstract. Please note that some parts.

 Please see Spano et al., 2019 (10.3389/fpsyg.2019.00368), Caffò et al., 2020 (10.3389/fpsyg.2020.00917) to improve the introduction of the ms.

Thank you for your suggestion. We have now studied this paper and we were surprised by the large scale of the Italian study. We would like to refer to it in the future and we hope to conduct a more detailed evaluation in Japan. We have added this relevant Italian research paper to the reference list and refered to it in the revised manuscript.

Main text, line 44-47

Please add the power analysis in order to compute the sample size, in the study population. Please move from the results to the study population section the final sample size, the characteristics of the sample and Figure 1.

The power analysis was conducted and the sample sizes were calculated to be 76 and 85 for the social environmental factors “inadequate public transport” and “no alternative driver,” respectively. The final sample size, the characteristics of the sample, and Figure 1 have been moved to Section 2.1 (Study population) in the revised manuscript.

Regarding the sample size in the logistic regression analysis, previous studies have shown that a small number of categories with a size 10 times greater than the explanatory variable is acceptable (e.g., Peduzzi et al. 1996). In this study, there were more than 80 non-resuming driving groups, i.e., 10 times more groups than the number of explanatory variables (there were 8 explanatory variables).

Please add the future direction of the study in the discussion section

Thank you for your suggestion. The revised manuscript contains a section on future prospects.

Main text, line 277-280

Round 2

Reviewer 1 Report

My thanks to the authors for their thorough reponses to my comments and queries. The article is much improved and I am happy for it to proceed.